# Direct costs of blood drawings with pre-analytical errors in tertiary paediatric hospital care

Henrik Hjelmgren[1,2]*, Emelie Heintz[3], Britt-Marie Ygge[1,2], Nina Andersson[1,2], Björn Nordlund[1,2]

1 Astrid Lindgren Children's Hospital, Karolinska University Hospital, Stockholm, Sweden, 2 Department of Women's and Children's Health, Karolinska Institute, Stockholm, Sweden, 3 Department of Learning, Informatics, Management and Ethics, Karolinska Institute, Stockholm, Sweden

* Henrik.hjelmgren@ki.se

**Data Availability Statement:** All relevant data are within the paper and its Supporting information files.

## Abstract

### Background

Blood drawings is a common hospital procedure involving laboratory and clinical disciplines that is important for the diagnosis and management of illnesses in children. Blood drawings with pre-analytical error (PAE) can lead to increased costs for hospitals and healthcare organisations. The direct cost of blood drawings after a PAE is not fully understood in paediatric hospital care.

### Aim

The aim of this study was to estimate the average direct cost of PAE per year and per 10,000 blood drawings in tertiary paediatric care.

### Methods

A cost analysis using a bottom-up approach was conducted on the basis of combined information from the hospital's laboratory register for the period 2013–2014 and clinical in-ward observations at a tertiary children's referral hospital in Sweden, the Astrid Lindgren Children's Hospital. For the analysis, we hypothesised the re-collection of all blood drawings with PAE and included the average costs of the sampling materials, the time of the healthcare personnel, the laboratory analyses, and in-ward premises based on the time spent on the blood sampling procedure.

### Results

The annual cost of PAE was estimated to be 74,267 euros per 54,040 blood drawings, which corresponds to 13,756 euros per 10,000 blood drawings or 1.5 euros per draw. The personnel cost represented 60.1% (45,261 euros per year) of the cost due to PAE, followed by costs for hospitalisation (25.2%), laboratory analyses (8.1%), and materials (5.7%).

**Funding:** The author(s) received no specific funding for this work.

**Competing interests:** The authors have declared that no competing interests exist.

## Conclusion

PAEs lead to substantial increases in the costs in tertiary paediatric hospital care. If these PAEs can be avoided, costs related to the re-collection of blood drawings with PAE may be re-allocated to other health-promoting activities for children visiting hospital institutions.

## Introduction

Blood drawings is important for the diagnosis and management of illnesses in children. It involves many stakeholders in laboratory and clinical settings [1]. Therefore, blood sampling errors cause many problems for both patients and healthcare workers that result in the need for repeated blood drawings and delayed decisions in paediatric health care [2, 3]. From a laboratory perspective, the total testing process can be divided into three phases: pre-analytical, analytical, and post-analytical. Approximately 70% of errors occur in the pre-analytical phase [4]. Pre-analytical errors (PAE) are defined as clotted or haemolysed blood samples, test tubes incorrectly filled, incorrect sample type, test transcription errors, unsuitable samples for transportation, storage problems, and mis-identification errors [5]. PAE can lead to severe health issues such as inappropriate treatment or even misdiagnosis, and delayed care [6–8]. Evidence shows that improvements in blood sampling procedures could support the quality and safety of healthcare services [9].

We previously reported a 5% prevalence of rejected laboratory blood analyses due to PAEs, especially due to clotted samples and test tubes which was filled incorrectly, in a paediatric tertiary hospital [10]. For hospital organisations, all laboratory costs represent approximately 5% of the total budget, but the diagnostic importance is extensive, as tests may influence 60–70% of all medical decisions [11]. Several studies have shown increased healthcare costs related to PAEs using different data sources that reflect personnel, material, and analytical outcomes [12–14]. Furthermore, other variables such as PAE cost in relation to a specific blood analysis [15] have been studied, but only a few studies have evaluated the direct costs of PAE in paediatric care. Therefore, the aim of this study was to estimate the average direct cost of blood drawings with PAE per year and per 10,000 blood drawings in tertiary paediatric hospital care.

## Materials and methods

### Design and setting

This pragmatic bottom-up approach cost analysis study of annual blood drawings affected by PAEs combined information collected at three different time points from the hospital's laboratory information system (FlexLab), hospital economic information system (Tableau software), hospital supply system (Medicarrier AB), and clinical observations. The study was performed at a tertiary paediatric hospital in Stockholm, Sweden, the Karolinska University Hospital-Astrid Lindgren Children's Hospital, which had the capacity of 129 beds in 2019. The hospital provides regional and national health care for children 0–18 years of age. The care provided to approximately 13,000 children yearly includes various specialities, namely surgery, medicine, oncology, neonatology, and intensive care.

**Direct cost analysis of the blood drawing process.** A bottom-up approach was used in this study to estimate the direct PAE costs per year and per 10,000 blood drawings at the Astrid Lindgren Children's Hospital. Thus, the cost analysis approach covered the following steps in the blood sampling process: first, the healthcare resources used in the blood drawing process from the pre- to the post-analytical phase were measured and quantified (Fig 1), and second,

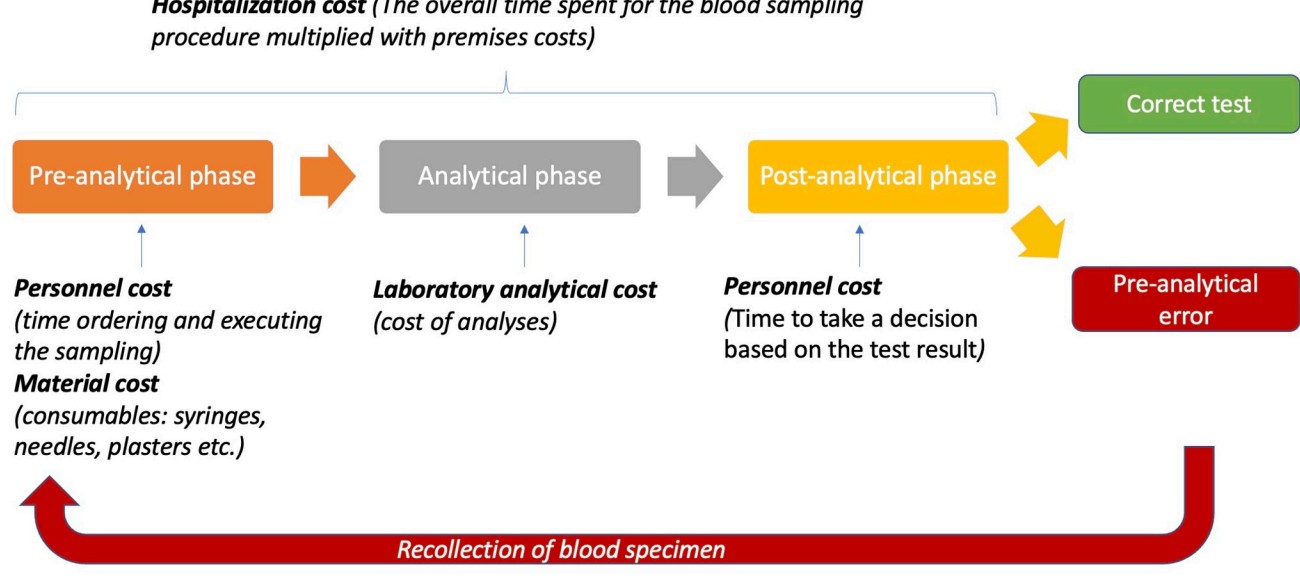

**Fig 1. Cost of the resources used in the blood drawing process.**

the identified quantified resources used were valued by multiplying the quantities with the corresponding unit costs [16, 17]. The assumption of this study analysis was that each blood draw with PAE leads to re-collection, which will consequently affect the direct costs of each step in the blood sampling process. The direct healthcare costs were considered to be related to the additional time for the re-collection of blood samples, with cost-associated effects on healthcare personnel, materials, laboratory analyses, and hospital in-ward premises.

*Personnel costs.* The time that healthcare personnel (medical doctors, registered nurses, and nurse assistants) spend on blood drawings was estimated on the basis of clinical observations at the emergency department and the neurology/orthopaedic ward in May 2021. The observations included the timing of the procedures, as outlined in Fig 2. The total allocated time for the blood sampling procedures was documented in relation to the healthcare profession and multiplied by the average hourly salary of medical doctors, registered nurses, and nurse assistants (S4 Table, salaries). In Swedish paediatric hospital settings, the blood tests are ordered by medical doctors and the drawings performed by registered nurses together with nurse assistants. An observation protocol was used to keep track of the time used by each professional. A total of 17 observations were made, and 15 registered nurses and 12 nurse assistants were included (S3 Table, observation protocol). The time medical doctors spent in the blood drawing procedure was estimated based on three medical doctors reported the average time they spend on ordering blood tests and reviewing the results in the electronic health record. Information on the average salary per hour/profession was collected from the hospital economic

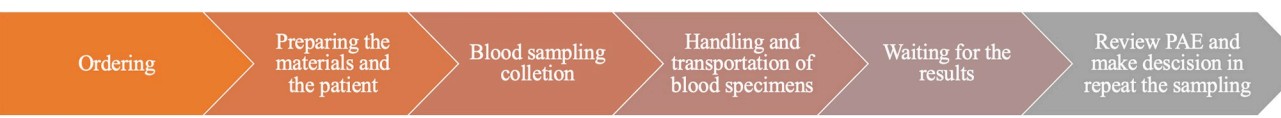

**Fig 2. Work procedures of healthcare personnel in the blood drawing process.**

information system (Tableau software). The professional cost was added to the cost of the time spent on the total blood sampling procedure. The details are summarised in Table 1.

*Material cost*. The cost of the materials used was estimated on the basis of the clinical observations of one paediatric nurse with pre-specified knowledge concerning blood drawings. The costs of the materials used for capillary and venous blood sampling are described in Table 1. The unit costs for consumables were collected from the hospital supply system and price lists for 2020 (Medicarrier AB, Stockholm Sweden; S1 Table). In our previous study, we investigated the PAE rate between venous and capillary blood samplings. Data from that study were used to estimate the proportion of materials used for the total data set [18].

*Laboratory cost*. At the Karolinska University laboratory, approximately 54,040 blood drawings were recorded from the Astrid Lindgren Children's Hospital and analysed at the coagulation, haematology, and chemistry laboratory sections. In total, these sections have analysed approximately 570,000 blood analyses per year [10]. For the analytical phase, the study covered the laboratory costs for the most common blood analyses in the biochemistry laboratory at Karolinska University Hospital, Sweden. The prices of these analyses were collected from the hospital laboratory price list for 2019 (S2 Table). Rare and unusually expensive blood analyses were excluded from the cost analysis by including only blood analyses requested at least 1,000 times per year in the cost analysis. The average cost per blood analyses was estimated by dividing the total cost of all blood analyses by the total number of blood analyses.

*Hospitalisation cost*. The hospitalisation cost was based on the overall time spent for the re-collection of a blood drawing due to a PAE, multiplied to the costs of in-ward premises. The

**Table 1. The direct costs for blood drawings at the Astrid Lindgren Children's Hospital is based on added up the costs model including personnel costs, used materials, average laboratory analyses costs, and duration of the use of premises (hospitalisation costs).**

| Resource | | Unit cost per blood drawing (euro) | Total cost per blood drawing (euros) |
|---|---|---|---|
| **Personnel costs** | Time spent | | |
| Medical doctor | 5 min | 39.22/h | 3.27 |
| Registered nurse | 21.3 min | 22.77/h | 8.08 |
| Nurse assistant | 15 min | 16.63/h | 4.16 |
| Summary of personnel cost per drawing | | | **15.51** |
| **Material costs** | Proportion of materials used* | | |
| Venous sampling using the open-needle technique | 2.6% | 4.17 | 0.11 |
| Venous sampling by peripheral vein catheter (PVC) draw with new insertion | 11.3% | 4.58 | 0.52 |
| Venous sampling by PVC draw | 23.7% | 0.87 | 0.21 |
| Venous sampling using the butterfly needle technique | 3.1% | 4.07 | 0.13 |
| Venous sampling using a straight-needle Vacutainer | 1.6% | 3.55 | 0.06 |
| Venous sampling by drawing from the central lines | 20.6% | 1.41 | 0.29 |
| Capillary sampling by finger prick | 35.3% | 0.38 | 0.14 |
| Capillary sampling by side-of-heel prick | 1.89% | 1.42 | 0.03 |
| Summary of material cost per drawing | | | **1.46** |
| **Average laboratory cost** | Per analysis | | |
| Laboratory analysis cost | 1 | 2.06 | **2.06** |
| **Hospitalisation cost** | Average time | | |
| Premises | 82.7 min | 112.7 | **6.42** |
| **Total cost of resources** | | | **25.45€** |

*The frequency of the different blood sampling methods performed in-ward at the Astrid Lindgren Children's Hospital, as published elsewhere [18].

blood sampling phases from start to end are described in Fig 1. The timesheet of the recorded observations is summarised in S3 Table. The cost of the premises used for hospitalisation during blood drawing was €112.7 per patient per day (only including the facilitating costs with a 2% overhead) for the general paediatric and neonatal wards, according to the hospital's economic information system (Tableau software).

**Blood drawings with PAEs.**    Descriptive statistics were used to illustrate the costs of the total testing process in relation to PAEs for the mentioned variables and outcomes. Information about the frequency of PAE was retrieved from the Karolinska University Hospital laboratory information system (FlexLab) between 2013 and 2014 [10]. PAE was defined as the rejection of a blood test result by the laboratory because of errors such as haemolysis, clotting, unfilled or wrong tubes, missing samples, and transportation errors [8]. The direct costs of personnel, materials, laboratory analyses, and hospitalisations are summarised and separately calculated in Table 1. The prevalence of PAEs was 5.4% (5.6% in 2013 and 5.2% in 2014) [10]. To generalise the cost of PAEs to hospitals where other PAEs are prevalent, a one-way sensitivity analysis was performed to illustrate the costs per 10,000 blood drawings related to one percentage change in the frequency of PAE. The currency was converted from Swedish kronor to euro at an exchange rate of 1 SEK = 0.0945 euro (€) in 2019.

### Ethical considerations

The ethical application of this study was exempted from ethical review by opinion of the Swedish Ethical Review Authority (DNR 2021–00846). The exemption of ethical review was due to the study do not involve any processing of sensitive personal data, therefore, is the study of such nature that it is not covered by the Swedish Ethical Review Act (2003:460). Nevertheless, in mind of ethical considerations, the researchers of this study have taken relevant precautions according to regulations of The Declaration of Helsinki. Written informed consent from the medical director of the ward was collected before the observation. Healthcare personnel also gave their consent at the time they received information about the study and the purpose of the observation. With respect to the integrity of the children and their parents, the observer was not present in the room during the blood drawing procedure. The original study evaluating the prevalence of PAEs at Astrid Lindgren Children's Hospital [10] was approved by the Swedish Ethical Review Authority (original registration number: 2015/206-31/4).

### Results

The results demonstrated that the average cost per blood draw was 25.4 euros, which includes the costs for personnel (11.4 euros), materials (1.5 euros), laboratory analysis (2.1 euros), and hospitalisation (6.4 euros; Table 2). The annual estimated cost of PAEs in the Astrid Lindgren

**Table 2.  Average annual healthcare costs of blood drawings with PAEs in a tertiary paediatric hospital.**

|  | Cost per blood draw, (€) | Annual cost per 54,040 blood drawings, (€) | Cost due to PAEs (frequency 5.4% *), (€) | Cost proportion, (%) |
|---|---|---|---|---|
| Personnel cost | 15.5 | 838,160 | 45,261 | 60.1 |
| Material cost | 1.5 | 78,898 | 4,261 | 5.7 |
| Laboratory analysis cost | 2.1 | 111,322 | 6,011 | 8.1 |
| Hospitalisation cost during blood drawings | 6.4 | 346,937 | 18,735 | 25.2 |
| **Total cost** | **25.4** | **1,375,317** | **74,267** | **100** |

*The frequency of PAEs based on FlexLab data (2013–2014).

**Table 3. Cost of PAEs per 10,000 blood drawings.**

| | Cost per 10,000 drawings (€) | Cost of PAEs per 10,000 drawings* (€) |
|---|---|---|
| Personnel cost | 155,100 | 8,375 |
| Material cost | 14,648 | 791 |
| Laboratory analysis cost | 20,790 | 1,123 |
| Hospitalisation cost during blood drawings | 64,200 | 3,467 |
| **Total cost** | **254,738** | **13,756** |

*The frequency (5.4%) of PAEs based on FlexLab data (2013–2014).

Children's Hospital was approximately 74,000 euros (at a 5.4% prevalence of PAEs in a 54,040 blood drawings) or 1.37 euro per blood draw in the hospital (Table 2). The time of the personnel represented the highest cost at approximately 45,000 euros per year, followed by the costs for hospitalisation at 19,000 euros, laboratory analyses at 6,000 euros, and materials used at 4,000 euros (Table 2).

The total annual average cost for all blood samples obtained in the hospital (54,040 blood drawings) was approximately 1.4 million euros, representing approximately 0.7% of our children's hospital's annual budget of approximately 195.1 million euros (year 2019; Tableau software).

The cost of PAEs per 10,000 blood drawings was estimated to be 13,756 euros (Table 3).

A sensitivity analysis was performed to illustrate that the direct cost of PAEs increases per percentage with 2,547 euros per 10,000 blood drawings (S1 Fig).

## Discussion

The aim of this study was to estimate the specific costs of re-collection of blood drawings due to PAEs in a tertiary paediatric hospital. The direct costs of PAEs were estimated to be 74,267 euros per 54,040 blood drawings and per year, or 13,756 euros per 10,000 blood drawings and per year.

The average direct costs of PAEs per year and per 10,000 blood drawings were higher than those in an adult emergency department based in Italy [13] and a tertiary care hospital in Canada [15], which spent approximately 1,174 euros (2014) and 3,275 euros (2013) per 10,000 sample collections, respectively. However, these values were lower than that reported in a German study in 2015 that estimated costs ranging from 34,000 to 61,000 euros per 10,000 adult sample collections [12]. The study from Canada only investigated coagulation international normalised ratio through blood analyses, which explains the low cost [15]. In Italy and Germany, the mean time for the registered nurses to execute an adult venepuncture was 2.5 and 10 min, respectively, which are very short in relation to the time required for paediatric samplings in our study, which was estimated to be 21,3 min for registered nurses. Furthermore, the studies from Italy, Canada, and Germany did not include the cost of the time of medical doctors and nurse assistants, which could have resulted in higher costs in our paediatric sampling, as we observed it to be 5 min for medical doctors and 15 min for nurse assistants. Even though the abovementioned studies excluded hospitalisation costs, we argue that hospitalisation costs are an important estimate that must be considered in PAE cost calculations. Overall, paediatric hospital care could also be more expensive than general hospital care owing to the more complex type of health care [19]. Blood drawings are often more challenging, extensive and time-consuming procedure in children compared to adults [20, 21]. Small blood volumes

and problems with incorrected filled samples seems to be more common in paediatric hospital care, which increases the risk of PAE, e.g. for coagulation analyses [10]. Additionally, the paediatric phlebotomy guidelines highlight the need to use comfort techniques and interact with the child as well as to use assistant staff to reduce distress and pain during the procedure [22, 23]. Depending on which comfort methods the staff use, it may take longer or shorter time [24]. Another aspect is the difference in time spent between inpatient and outpatient blood drawings. The inpatient is often more severely ill and have often gone through repeated needle-related procedures, which makes it more challenging to calm down and prepare an inpatient-child for a blood drawing.

This study elucidates the importance of reducing the incidence rate of PAE in paediatric tertiary care to reduce unnecessary costs for the organisation. If the blood drawing procedure is performed correctly and thus avoids re-collection due to PAEs, this may release resources that can be used for other activities in hospital in-wards and emergency wards. For comparison, the average annual salary of two registered nurses in 2019 in Sweden was 80,000 euros. To our knowledge, the direct cost of PAEs in a paediatric hospital has been seldom analysed separately from adult hospital laboratory analysis reports. In this context, we believe our study makes an important contribution, as the blood drawing procedure differs significantly between paediatric and adult care in terms of locating the veins, lower blood sampling volume, longer preparations, and sometimes more stressful situations for both the child and the staff [22–26].

In our cost analysis, we assumed that all PAEs led to 100% re-collection. As far as we know, no studies have evaluated how often re-collection occurs due to PAEs in paediatric care. The adult context has shown that 86.6% of PAEs potentially lead to repeated sampling [2]. If this is similar in paediatric care, the associated PAE costs could be lower than indicated in our results. In paediatric wards, re-collections are not always executed to spare the children any additional punctures [27]. By contrast, it is sometimes necessary to repeat the blood sampling procedure more than once due to multiple PAEs. Future research will have to evaluate the best practices and interventions for paediatric care and reduce PAEs and the number of re-collections, thereby saving costs. The one-way sensitivity analysis of this study revealed that only a 1% decrease in the frequency of PAEs may reduce the PAE cost per 10,000 blood drawings by 2,500 euros. This indicates that the small measures that reduced the frequency of PAEs might have a large impact on the overall costs. The costs of PAE could be reduced by targeted interventions such as educational and technical solutions [28–30], but whether they are applicable in the paediatric context is unclear. Another possibility to reduce cost is the use of phlebotomy teams [31]. In the Swedish context, phlebotomists only work in outpatient clinics, and to the best of our knowledge, phlebotomy teams have not been investigated in a paediatric in-ward hospital setting.

## Methodological considerations

This cost analysis study presented the costs of PAE according to personnel, materials, and analytical direct costs, but adapted changes for the context of a paediatric hospital setting, with a study design similar to that of previous studies [12, 15, 28]. The effect of PAEs on hospitalisation costs could be questioned, as hospitalisation cost may be considered a fixed cost with limited association to the frequency of PAEs. Furthermore, we could not estimate the unpredicted indirect cost of the possible consequences of PAEs, such as an increased need for treatments, blood transfusion, antibiotics, radiographs, or rehabilitation. Green (2013) estimated that annually, approximately 16,000 patient hours are lost because of PAEs, which leads to the re-draw of samples and to an additional cost for patient treatment, ranging from

178–245 euros per PAE [32]. This is much higher than the result of this study, which was 1.37 euros per PAE. In a Swedish paediatric hospital context, consulting an anaesthesia nurse is sometimes needed because of the difficulties of needle-related procedures, which could also lead to increased personnel costs. The long-term effects of children with needle phobia due to excessive blood sampling also have potential costs at the societal level and costs for consultation with play therapists for distraction and rehabilitation for post-traumatic experiences [33].

The included costs in this study were also specified only when specimens were sent to the biochemistry section of the laboratory. Other blood test analyses such as microbiological, pathological, immunological, and point-of-care testing analyses were not included. This means that PAE-related costs could be even higher if all laboratory sections receiving blood tests from paediatric patients were included.

A limitation of this study was that the observations were made by a single observer, in this case, the first author (HH). The collected data might have deviated because of the conscious or subconscious mindset of the observer [34]. To mitigate information biases, an observation protocol and documented memos were used. Adding a blind observer to the exposed status may reduce the incidence of differential information errors, but this was not possible for practical reasons within the scope of the present study.

## Conclusion

This cost analysis study estimated blood drawing re-collection costs due to PAEs and indicated that the annual cost of PAEs was 74,267 euros or 13,756€ per 10,000 blood drawings in tertiary paediatric hospitals. Among the cost categories included in the costs of PAEs, personnel costs represented the highest cost. Reducing the incidence of PAEs could yield significant cost reductions.

## Supporting information

**S1 Table.** **a**: Material consumable cost at the Astrid Lindgren's Children's Hospital (2020). **b**: Material consumables cost and unit use per blood drawing method at Astrid Lindgren's Children's Hospital (2020).
(DOCX)

**S2 Table. The analysis cost of blood tests ordered and analysed in the chemistry, haematology and coagulation sections at Astrid Children's Lindgren's Hospital 2019.**
(DOCX)

**S3 Table. Observation clinical protocol of 17 blood drawings (7 capillary and 10 venous) in May 2021 at the Astrid Lindgren's Children's Hospital.**
(DOCX)

**S4 Table. Personnel salaries (2019) from the health information system (Tableau software).**
(DOCX)

**S1 Fig. Direct costs of PAEs (€) per 10,000 blood drawings according to the prevalence rate of PAEs from 0% to 10%.**
(TIFF)

## Acknowledgments

The authors sincerely appreciate the administration staff at the Astrid Lindgren's Children's Hospital, namely Ann Brynjer, Mats Karlsson, and Ulrika Holmén, who helped us retrieve data from different resources, making the analysis of this study possible.

## Author Contributions

**Conceptualization:** Henrik Hjelmgren, Britt-Marie Ygge, Nina Andersson, Björn Nordlund.

**Methodology:** Emelie Heintz, Björn Nordlund.

**Supervision:** Björn Nordlund.

**Writing – original draft:** Henrik Hjelmgren, Emelie Heintz, Björn Nordlund.

**Writing – review & editing:** Britt-Marie Ygge, Nina Andersson, Björn Nordlund.

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
