## [Decision Letter · Decision Letter 0]

4 Jan 2023

PONE-D-22-29412Direct costs associated with failed blood sample collections in tertiary paediatric hospital care

PLOS ONE

Dear Dr. Hjelmgren,

Thank you for submitting your manuscript to PLOS ONE. After careful consideration, we feel that it has merit but does not fully meet PLOS ONE’s publication criteria as it currently stands. Therefore, we invite you to submit a revised version of the manuscript that addresses the points raised during the review process.

The manuscript has been evaluated by three reviewers, and their comments are available below.

The reviewers have raised concerns regarding the reporting, methodology and conceptualization of this study. 

Could you please revise the manuscript to carefully address the concerns raised?

We look forward to receiving your revised manuscript.

Kind regards,

Johannes Stortz, PhD

Staff Editor

PLOS ONE

Journal Requirements:

3. You indicated that ethical approval was not necessary for your study. We understand that the framework for ethical oversight requirements for studies of this type may differ depending on the setting and we would appreciate some further clarification regarding your research. Could you please provide further details on why your study is exempt from the need for approval and confirmation from your institutional review board or research ethics committee (e.g., in the form of a letter or email correspondence) that ethics review was not necessary for this study? Please include a copy of the correspondence as an "Other" file.

6. We noted in your submission details that a portion of your manuscript may have been presented or published elsewhere. Please clarify whether this publication was peer-reviewed and formally published. If this work was previously peer-reviewed and published, in the cover letter please provide the reason that this work does not constitute dual publication and should be included in the current manuscript.

Reviewers' comments:

Reviewer's Responses to Questions

**Comments to the Author**

1. Is the manuscript technically sound, and do the data support the conclusions?

Reviewer #1: Partly

Reviewer #2: Yes

Reviewer #3: No

2. Has the statistical analysis been performed appropriately and rigorously? 

Reviewer #1: N/A

Reviewer #2: Yes

Reviewer #3: N/A

3. Have the authors made all data underlying the findings in their manuscript fully available?

Reviewer #1: Yes

Reviewer #2: Yes

Reviewer #3: Yes

4. Is the manuscript presented in an intelligible fashion and written in standard English?

Reviewer #1: Yes

Reviewer #2: Yes

Reviewer #3: Yes

5. Review Comments to the Author

Reviewer #1: The authors present important data for cost estimation of preanalytical errors, which could serve as basis for negotiations with hospital management, aiming to improve quality of care.

The authors emphasise that their study is the first one focusing on paediatric care.

If focusing only on children, the PAE rate from the laboratory has to be recalculated, including only samples from pediatric care.

How was the total allocated time to the blood sampling procedures calculated/observed? If observations were done, how many per profession?

30 minutes for blood collection for a nurse seems quite long, especially when including 15 minutes nurse assistant and 5 min doctors time

Why are personell costs added up? Same for material costs. I believe that not all types of needles are used simultaneously for each blood collection

Fig 3 needs further explanation as well as an x-axis title. The figure needs to be understandable only by the image and the legend.

Reviewer #2: Occasional grammatical errors

The paper should state at the outset that the paper is an analysis of data collected at different times (PAE data from 2013/2014, finance data from other years etc). This is not unreasonable but should be described as the structure for the work.

The procedure for the collection of data for personnel cost etc by observation should be described in the methods. The discussion mentions the use of standard protocols to reduce bias and this should be in the methods.

The 'hospitalisation cost' is unclear. I assumed this was the overheads costs (building, utilities, service costs etc) but the figure attached includes the steps of the pre, analytical and post phases.

Table 1 would benefit from clarification - the first column (allocation per blood sampling) has data in minutes, % etc. with little explanation.

The discussion reviews data from other papers and attempts to explain the variations. The overall variation between papers is enormous=: this may be because the data has been collected so differently that it is not comparable. I wonder if there are any components between the different studies that can be normalised to make them comparable?

The histogram figure presented is difficult to understand, even when referring to the legend in the text. It requires revising to contribute to the paper. In fact, all 3 figures would benefit from refining.

Reviewer #3: This manuscript describes a calculation over the costs associated with blood sample collections that due to preanalytical errors must be repeated. The study is based on register data from 2013-2014 and the data on the preanalytical errors has previously been published. Although I find it relevant to calculate the expenses caused by errors in the blood sampling, I do not find the study merits a scientific report. The novelty is too meager, and the basic findings are already published, If any, these calculations should have been a part of the first paper. Also, many things have changed in awareness on the preanalytical area in the last ten years, especially in the pediatric field.

Apart from this major remark, I have a number of minor remarks:

I do not understand why there are no phlebotomists involved in the blood sampling: In many countries, a subgroup of phlebotomists are specially trained in pediatric sampling, which has been shown to reduce the number of preanalytical errors. The authors should comment on this.

When the numbers of blood samplings are counted, there is a confusion between the blood sampling process and the number of samples. I would imagine that the 50,040 samples drawn are from are far lesser number of blood samplings; in general, 3-5 samples are taken per drawing, and the number of drawings are therefore only around 15,000. This alters the calculations, which are made per blood sample, which probably more correctly depicts a price per sampling (and 3-5 times more blood samples).

When calculating the prices, I wonder were costs for blood tubes are incorporated. Please specify.

In the original paper from 2019, it is shown that the top 2 in preanalytical errors are clotted samples (51,3%) and incorrectly filled samples (23,3%). The latter is actually only a real problem in terms of coagulation testing, and the pricing therefore depends on the analysis requested, as probably only few needs a re-draw.

I find that Figure 1-3 is irrelevant and can be omitted.

6. PLOS authors have the option to publish the peer review history of their article (what does this mean?). If published, this will include your full peer review and any attached files.

Reviewer #1: No

Reviewer #2: No

Reviewer #3: No

---

## [Author Response · Author response to Decision Letter 0]

8 Mar 2023

Dear Editor,

Thank you for letting us revise our manuscript ”Direct costs associated with failed blood sample collections in tertiary paediatric hospital care” for publication in PLOS ONE. 

We have answered point by point the questions aroused from the original review.

Journal Requirements:

https://journals.plos.org/plosone/s/file?id=wjVg/PLOSOne_formatting_sample_main_body.pdfand

 - Reply: Thanks. We have revised accordingly. 

- Reply: Thank you for pointing this out. We have added details regarding participants consent information in the manuscript under ethical consideration. 

3. You indicated that ethical approval was not necessary for your study. We understand that the framework for ethical oversight requirements for studies of this type may differ depending on the setting and we would appreciate some further clarification regarding your research. Could you please provide further details on why your study is exempt from the need for approval and confirmation from your institutional review board or research ethics committee (e.g., in the form of a letter or email correspondence) that ethics review was not necessary for this study? Please include a copy of the correspondence as an "Other" file.

- Reply: Thank you for this question. The study was exempted from ethical review due to the following quote from the ethical committee. “The research referred to in the application for change does not involve any processing of sensitive personal data. Nor is the research otherwise of such a nature that it is covered by the Act (2003:460) on ethical review of research involving humans. The ethics review authority cannot therefore take up the application for change for consideration for approval.”

Nevertheless, we as researchers have taken ethical considerations in mind in all this study process according to regulations in the Helsinki declaration.” 

- Reply: The original data set from Flexlab cannot be shared as it carries personnel data, which is regulated by our the ethical approval (DNR 2021-00846), see under Ethical Considerations, page 8. However, the observation protocol can be published, see supporting information files 3.

- Reply: Thank you. We believe we answered this above. 

6. We noted in your submission details that a portion of your manuscript may have been presented or published elsewhere. Please clarify whether this publication was peer-reviewed and formally published. If this work was previously peer-reviewed and published, in the cover letter please provide the reason that this work does not constitute dual publication and should be included in the current manuscript.

- Reply: This study has not been published in a scientific paper elsewhere, but data from the study is included in corresponding author Henrik Hjelmgren’s doctoral thesis, defended in February 2022 and published in open archive at Karolinska Institutet https://openarchive.ki.se/xmlui/bitstream/handle/10616/47834/Thesis_Henrik_Hjelmgren.pdf?sequence=1&isAllowed=y

- Reply: Thank you. We revised accordingly. 

Comments to the Author

Reviewer #1: 

The authors present important data for cost estimation of preanalytical errors, which could serve as basis for negotiations with hospital management, aiming to improve quality of care.

The authors emphasise that their study is the first one focusing on paediatric care.

If focusing only on children, the PAE rate from the laboratory has to be recalculated, including only samples from pediatric care.

- Reply: Thank you for your comment. The PAE rate includes only samples from pediatric care. Please see further information in “Hjelmgren H, et al. Retrospective study showed that blood sampling errors risked children's well-being and safety in a Swedish paediatric tertiary care. Acta Paediatr, 2019. 108(3): p. 522-528”

How was the total allocated time to the blood sampling procedures calculated/observed? If observations were done, how many per profession?

- Reply: Thank you for your question. The total allocated time which was observed and calculated are presented in the timesheet which is summarised in S3 Table and now added details in S4. We made 17 observations whereby 15nurses and 12 nurse assistants were included. 10 times there were both a nurse and a nurse assistant. Just a few medical doctors were observed in the beginning of the study to estimate the average 5min time which was not documented in the protocol but with memos. We clarified this and added in the method section of the manuscript under personnel cost. 

30 minutes for blood collection for a nurse seems quite long, especially when including 15 minutes nurse assistant and 5 min doctors time

- Reply: Thank you for pointing this out. The observations included preparations, taking the sample, as well as the things you need to do after an sampling. We added to Supporting information 3 a more detailed observation protocol to be able to clarify for the reader our time observations. It’s possible so that inward hospital settings the time can take longer than outpatient clinics as the HCPs has many different tasks going on at the same time. 

Why are personell costs added up? Same for material costs. I believe that not all types of needles are used simultaneously for each blood collection

- Reply: Thank you for this question. To align with to use of different materials for blood sampling the costs were added up based on the study “Capillary blood sampling increases the risk of preanalytical errors in pediatric hospital care: Observational clinical study” showing the prevalence of different blood sampling methods inward at Astrid Lindgren Children’s Hospital. The added up personal costs reflects the different healthcare personnel’s inward that are involved in the blood sampling process, see revised Table 1. 

Fig 3 needs further explanation as well as an x-axis title. The figure needs to be understandable only by the image and the legend.

- Reply: Thank you for pointing this out. We have clarified the x-axis title and put the figure 3 to supplementary information 4 instead. 

Reviewer #2: Occasional grammatical errors

The paper should state at the outset that the paper is an analysis of data collected at different times (PAE data from 2013/2014, finance data from other years etc). This is not unreasonable but should be described as the structure for the work.

- Reply: Thank you for this suggestion. We have added this to the method/design section at page 4. 

The procedure for the collection of data for personnel cost etc by observation should be described in the methods. 

- Reply: Thank you for this comment. We revised and clarified in the manuscript under personnel cost in the methods section. 

The discussion mentions the use of standard protocols to reduce bias and this should be in the methods.

- Reply: Thank you for this comment. It was not a “standard” protocol it was an observation protocol adjusted for this purpose only. We have now added a more detailed information in the S3 table - observation protocol. We also clarified the same information under the method section personnel cost. 

The 'hospitalisation cost' is unclear. I assumed this was the overheads costs (building, utilities, service costs etc) but the figure attached includes the steps of the pre, analytical and post phases.

- Reply: Thank you for mention this point. The hospitalization cost is as you say overhead costs but only during the time sampling occurred (pre to-post phases). We revised and made a clarification under this method section page 7 and row 151.

Table 1 would benefit from clarification - the first column (allocation per blood sampling) has data in minutes, % etc. with little explanation.

- Reply: Thanks for your comment, Table 1 has been revised accordingly. 

The discussion reviews data from other papers and attempts to explain the variations. The overall variation between papers is enormous=: this may be because the data has been collected so differently that it is not comparable. I wonder if there are any components between the different studies that can be normalised to make them comparable?

- Reply: Thank you for this comment. We totally agree with you and it was a stress on how to attempt to refer to other publications in this field. Our attempt was to normalize and compare with the frequency 10,000samplings and by converting and summarize the cost in euros. 

The histogram figure presented is difficult to understand, even when referring to the legend in the text. It requires revising to contribute to the paper. In fact, all 3 figures would benefit from refining.

- Reply: Thank you for this point. We refined Fig 1 and Fig 3. We moved Fig 3 to Supplementary. 

Reviewer #3: This manuscript describes a calculation over the costs associated with blood sample collections that due to preanalytical errors must be repeated. The study is based on register data from 2013-2014 and the data on the preanalytical errors has previously been published. Although I find it relevant to calculate the expenses caused by errors in the blood sampling, I do not find the study merits a scientific report. The novelty is too meager, and the basic findings are already published, If any, these calculations should have been a part of the first paper. Also, many things have changed in awareness on the preanalytical area in the last ten years, especially in the pediatric field. 

-Reply: Thank you for your remark. To the best of our knowledge we believe that there is novelty and scientific awareness in the study which could be of interest to the readers of PLOS1 as well as clinicians around the world. 

Apart from this major remark, I have a number of minor remarks:

I do not understand why there are no phlebotomists involved in the blood sampling: In many countries, a subgroup of phlebotomists are specially trained in pediatric sampling, which has been shown to reduce the number of preanalytical errors. The authors should comment on this.

- Reply: Thank you for pointing this out. This is the case in some countries as you mention but not in the Swedish context. Especially when it comes to inward and hospital settings phlebotomists are rare. In the paediatric field there are several advantages if the staff know their patients and has knowledge on children’s background, different development stages as well as medical conditions. This is seldom anything phlebotomist have time and knowledge in which is something to debate of course. To our knowledge we haven’t seen any studies/publication comparing peadiatric draws by phlebotomist and peadiatric draws by peadiatric nurses with the outcome of PAE and children’s satisfaction and costs. We added in the manuscript discussion part Page 16, row 255: 

“Another possibility to reduce cost is the use of phlebotomy teams [29]. In the Swedish context, phlebotomists only work in outpatient clinics, and to the best of our knowledge, phlebotomy teams have not been investigated in a paediatric in-ward hospital setting.

When the numbers of blood samplings are counted, there is a confusion between the blood sampling process and the number of samples. I would imagine that the 50,040 samples drawn are from are far lesser number of blood samplings; in general, 3-5 samples are taken per drawing, and the number of drawings are therefore only around 15,000. This alters the calculations, which are made per blood sample, which probably more correctly depicts a price per sampling (and 3-5 times more blood samples).

- Reply: Sorry for the misunderstanding. While the number of specific blood analyses were 1,148,716, for the two years 2013-2014, the number of unique blood samplings were 108,080. We then calculated the annual mean to 54,040 (Hjelmgren et al 2019). 

When calculating the prices, I wonder were costs for blood tubes are incorporated. Please specify.

- Reply: Thank you for this question. This was incorporated in the material cost of each sampling method depending on the method. Supplementary table 1 shows the different tubes cost which was used. 

In the original paper from 2019, it is shown that the top 2 in preanalytical errors are clotted samples (51,3%) and incorrectly filled samples (23,3%). The latter is actually only a real problem in terms of coagulation testing, and the pricing therefore depends on the analysis requested, as probably only few needs a re-draw. 

- Reply: Thank you for this comment, however we don’t agree. Incorrect filled samples can occur in all different testings. In our study from 2019 we found the number of incorrect filled samples in 7494 analyses in the hematology section and 1225 analyses in the coagulation and 4230 analyses in the chemistry section. But by proportion of the blood analyses taken, coagulation had 2% and hematology and chemistry had 1% respectively. Due to importance of the clinical value of coagulation testing this is probably to most common re-draw analysis in paeditrics as the filling is to be on the exact marking on the tube as well. 

I find that Figure 1-3 is irrelevant and can be omitted.

- Reply: ok. Thank you for the suggestion. We omitted figure 3 to supplementary information but kept figure 1- 2 in the manuscript but made small adjustments. 

6. PLOS authors have the option to publish the peer review history of their article (what does this mean?). If published, this will include your full peer review and any attached files.

Do you want your identity to be public for this peer review? For information about this choice, including consent withdrawal, please see our Privacy Policy.

Reviewer #1: No

Reviewer #2: No

Reviewer #3: No

---

## [Decision Letter · Decision Letter 1]

14 Jun 2023

PONE-D-22-29412R1Direct costs associated with failed blood sample collections in tertiary paediatric hospital carePLOS ONE

Dear Dr. Hjelmgren,

Thank you for submitting your manuscript to PLOS ONE. After careful consideration, we feel that it has merit but does not fully meet PLOS ONE’s publication criteria as it currently stands. Therefore, we invite you to submit a revised version of the manuscript that addresses the points raised during the review process.

We look forward to receiving your revised manuscript.

Kind regards,

Janne Cadamuro

Guest Editor

PLOS ONE

Journal Requirements:

Additional Editor Comments:

Dear authors,

There still seem to be some major issues:

1) The time spent on blood collection is still an open issue. You point out that in supporting information 3a the rational for using 30 minutes as mean time for blood collections is 30 min. However, the numbers in that document read 21,3min. Additionally, as pointed out by reviewer #1, the time seems quite long compared to other data sources. See https://news.mayocliniclabs.com/2018/03/01/staffing-workload-phlebotomy-areas-direct-effort/

2) The rationale for adding up costs of different professions still is not convincing.

Please explain in the text and compare to other sources so the reader can objectively interpret your numbers compared to others.

Reviewers' comments:

Reviewer's Responses to Questions

**Comments to the Author**

1. If the authors have adequately addressed your comments raised in a previous round of review and you feel that this manuscript is now acceptable for publication, you may indicate that here to bypass the “Comments to the Author” section, enter your conflict of interest statement in the “Confidential to Editor” section, and submit your "Accept" recommendation.

Reviewer #2: All comments have been addressed

Reviewer #3: All comments have been addressed

2. Is the manuscript technically sound, and do the data support the conclusions?

Reviewer #2: Yes

Reviewer #3: Partly

3. Has the statistical analysis been performed appropriately and rigorously? 

Reviewer #2: Yes

Reviewer #3: N/A

4. Have the authors made all data underlying the findings in their manuscript fully available?

Reviewer #2: Yes

Reviewer #3: (No Response)

5. Is the manuscript presented in an intelligible fashion and written in standard English?

Reviewer #2: Yes

Reviewer #3: Yes

6. Review Comments to the Author

Reviewer #2: Thank you for addressing my comments on the first version. I have no further comments on the manuscript

Reviewer #3: I find the replies satisfying.

I do however still find that the terms blood samples and blood samplings (= drawings) are used incorrectly: The authors explain that the 54,040 refers to unique blood samplings. This is however not what the abstract says: “The annual cost of PAE was estimated to be 84,000 euros per 54,040 blood samples, which corresponds to 15,500 euros per 10,000 samples or 1.5 euros per sample.” I presume this is the cost per drawing? Please be more specific and recalculate if necessary - cost per drawing and per sample (with tube costs etc.) should be divided.

Regarding the incorrect filling, I am well aware of the fact that incorrect filled samples can occur in all different “testings” – but the importance is not the same at all: The coagulation analyses will for certain be errorred if samples are underfilled, while most other measurements will not be affected.

7. PLOS authors have the option to publish the peer review history of their article (what does this mean?). If published, this will include your full peer review and any attached files.

Reviewer #2: No

Reviewer #3: No

---

## [Author Response · Author response to Decision Letter 1]

12 Jul 2023

RESPONSE LETTER #2

PONE-D-22-29412R1

Direct costs associated with failed blood sample collections in tertiary paediatric hospital care

PLOS ONE

Dear Dr. Hjelmgren,

Thank you for submitting your manuscript to PLOS ONE. After careful consideration, we feel that it has merit but does not fully meet PLOS ONE’s publication criteria as it currently stands. Therefore, we invite you to submit a revised version of the manuscript that addresses the points raised during the review process.

• A rebuttal letter that responds to each point raised by the academic editor and reviewer(s). You should upload this letter as a separate file labeled 'Response to Reviewers'. YES. 

• A marked-up copy of your manuscript that highlights changes made to the original version. You should upload this as a separate file labeled 'Revised Manuscript with Track Changes'. YES. 

• An unmarked version of your revised paper without tracked changes. You should upload this as a separate file labeled 'Manuscript'. YES. 

We look forward to receiving your revised manuscript. 

Kind regards,

Janne Cadamuro

Guest Editor

PLOS ONE

Journal Requirements:

Reply: 

Dear Editor Dr Janne Cadamuro and reviewers. We sincerely thank you for the opportunity to resubmit a revised and improved version of the manuscript. We are grateful for the time and efforts you and the reviewer spent to provide valuable feedback. We have considered all comments by answering the comments point by point below in red, and we have revised the manuscript accordingly. In addition, we have changed the original title “Direct costs associated with failed blood sample collection in tertiary paediatric hospital care” to, in our opinion more informative and balanced summary, “Direct costs of blood drawings with pre-analytical errors in tertiary paediatric hospital care”.

These are the changes in the new reference list: 

Number 20: Inserted reference: 

20. Piazza J, Merkel S, Neusius H, Murphy S, Gargaro J, Rothberg B, et al. It's not just a needlestick: exploring phlebotomists' knowledge, training, and use of comfort measures in pediatric care to improve the patient experience. J Appl Lab Med. 2019;3(5):847-56

This reference was deleted: 

Trotochaud K, Coleman JR, Krawiecki N, McCracken C. Moral distress in pediatric healthcare providers. J Pediatr Nurs. 2015;30(6):908-14.

New references was inserted in 22-24: 

22: CLSI. Collection of Diagnostic Venous Blood specimens. 7th ed. CLSI standard GP41. Wayne, PA,: Clinical and Laboratory Standards Institute, 2017. Appendix A: Difficult Collection A1: Venipuncture in Children. 

23. World Health Organization, (2010). WHO Guidelines on Drawing Blood: Best Practices in Phlebotomy, Paediatric and neonatal blood sampling.

24. Normandin PA, Benotti SA. Pediatric Phlebotomy: Taking the Bite Out of Dracula. 

J Emerg Nurs. 2018 Jul;44(4):427-429. doi: 10.1016/j.jen.2018.03.017.

Therefore, the references 25-32 was updated to 27-34 accordingly. 

Additional Editor Comments:

Dear authors,

There still seem to be some major issues:

1) The time spent on blood collection is still an open issue. You point out that in supporting information 3a the rational for using 30 minutes as mean time for blood collections is 30 min. However, the numbers in that document read 21,3min. 

Additionally, as pointed out by reviewer #1, the time seems quite long compared to other data sources. See https://news.mayocliniclabs.com/2018/03/01/staffing-workload-phlebotomy-areas-direct-effort/

Reply: 

Thank you for valuable comment. We have now revised the time spent on blood drawings to 21.3 min and recalculated the costs accordingly.

Thank you for sending the link from Mayo Clinics, we note much shorter time on blood drawings. We have noted that the time spent on blood sampling is longer in children compared with adults. A clear distinction between blood sampling in children and adults, also stated in paediatric phlebotomy guidelines from WHO, Karolinska University Hospital Astrid Lindgren’s Children’s Hospital and CLSI Standard GP41 7th Ed*, is the preparation of the child and the use of pain relief methods, this can probably explain one of the reasons of the deviation in time spent in blood drawings between paediatric and adult care. We have updated the discussion to cover this topic more thoroughly, page 13, line 229-239. 

*CLSI. Collection of Diagnostic Venous Blood specimens. 7th ed. CLSI standard GP41. Wayne, PA,: Clinical and Laboratory Standards Institute, 2017. Appendix A: Difficult Collection A1: Venipuncture in Children. 

2) The rationale for adding up costs of different professions still is not convincing.

Please explain in the text and compare to other sources so the reader can objectively interpret your numbers compared to others.

Reply: Thank you for pointing this out. We have clarified the rationale for adding up costs in the method section, page 5, line 110-116. Please, see also the revised Table 1.

Reviewers’ comments:

Reviewer’s Responses to Questions 

Comments to the Author

1. If the authors have adequately addressed your comments raised in a previous round of review and you feel that this manuscript is now acceptable for publication, you may indicate that here to bypass the “Comments to the Author” section, enter your conflict of interest statement in the “Confidential to Editor” section, and submit your “Accept” recommendation.

Reviewer #2: All comments have been addressed

Reviewer #3: All comments have been addressed

2. Is the manuscript technically sound, and do the data support the conclusions?

Reviewer #2: Yes

Reviewer #3: Partly

3. Has the statistical analysis been performed appropriately and rigorously? 

Reviewer #2: Yes

Reviewer #3: N/A

4. Have the authors made all data underlying the findings in their manuscript fully available?

Reviewer #2: Yes

Reviewer #3: (No Response)

5. Is the manuscript presented in an intelligible fashion and written in standard English?

Reviewer #2: Yes

Reviewer #3: Yes

6. Review Comments to the Author

Reviewer #2: Thank you for addressing my comments on the first version. I have no further comments on the manuscript

Reviewer #3: I find the replies satisfying.

I do however still find that the terms blood samples and blood samplings (= drawings) are used incorrectly: The authors explain that the 54,040 refers to unique blood samplings. This is however not what the abstract says: “The annual cost of PAE was estimated to be 84,000 euros per 54,040 blood samples, which corresponds to 15,500 euros per 10,000 samples or 1.5 euros per sample.” I presume this is the cost per drawing? Please be more specific and recalculate if necessary – cost per drawing and per sample (with tube costs etc.) should be divided.

Reply: Thank you for pointing this out. We have revised accordingly and made it clearer in the text by changing to “drawings” instead of “samples/samplings” where it is appropriate. This has been done in the title, abstract and within the whole manuscript. 

The cost is estimated out of different ways of drawings when taking it venous or capillary. We added a Supplementary Table 1b showing the use of how the material was used for each method. 

Regarding the incorrect filling, I am well aware of the fact that incorrect filled samples can occur in all different “testings” – but the importance is not the same at all: The coagulation analyses will for certain be errorred if samples are underfilled, while most other measurements will not be affected.

Reply: Thank you once again for discussing this further. We agree with the point that coagulation errors are more sensitive for underfilled blood samples. This has now been noted in the Discussion section, page 13, line 231-233.

7. PLOS authors have the option to publish the peer review history of their article (what does this mean?). If published, this will include your full peer review and any attached files.

Do you want your identity to be public for this peer review? For information about this choice, including consent withdrawal, please see our Privacy Policy.

Reviewer #2: No

Reviewer #3: No

---

## [Decision Letter · Decision Letter 2]

13 Aug 2023

Direct costs of blood drawings with pre-analytical errors in tertiary paediatric hospital care

PONE-D-22-29412R2

Dear Dr. Hjelmgren,

We’re pleased to inform you that your manuscript has been judged scientifically suitable for publication and will be formally accepted for publication once it meets all outstanding technical requirements.

Kind regards,

Janne Cadamuro

Guest Editor

PLOS ONE

Additional Editor Comments (optional):

All comments have been addressed

Reviewers' comments:

Reviewer's Responses to Questions

**Comments to the Author**

1. If the authors have adequately addressed your comments raised in a previous round of review and you feel that this manuscript is now acceptable for publication, you may indicate that here to bypass the “Comments to the Author” section, enter your conflict of interest statement in the “Confidential to Editor” section, and submit your "Accept" recommendation.

Reviewer #2: All comments have been addressed

Reviewer #3: All comments have been addressed

2. Is the manuscript technically sound, and do the data support the conclusions?

Reviewer #2: Yes

Reviewer #3: Yes

3. Has the statistical analysis been performed appropriately and rigorously? 

Reviewer #2: Yes

Reviewer #3: N/A

4. Have the authors made all data underlying the findings in their manuscript fully available?

Reviewer #2: Yes

Reviewer #3: Yes

5. Is the manuscript presented in an intelligible fashion and written in standard English?

Reviewer #2: Yes

Reviewer #3: Yes

6. Review Comments to the Author

Reviewer #2: The authors have resubmitted the manuscript after two rounds of review comments from the reviewers and the guest editor. My comments have been addressed and I am satisfied that the authors have addressed the feedback from other reviewers.

Reviewer #3: (No Response)

7. PLOS authors have the option to publish the peer review history of their article (what does this mean?). If published, this will include your full peer review and any attached files.

Reviewer #2: No

Reviewer #3: No

---

## [Editor Report · Acceptance letter]

16 Aug 2023

PONE-D-22-29412R2 

Direct costs of blood drawings with pre-analytical errors in tertiary paediatric hospital care 

Dear Dr. Hjelmgren:

I'm pleased to inform you that your manuscript has been deemed suitable for publication in PLOS ONE. Congratulations! Your manuscript is now with our production department. 

Kind regards, 

on behalf of

Dr. Janne Cadamuro 

Guest Editor

PLOS ONE